# Current Promising Strategies against Antibiotic-Resistant Bacterial Infections

**DOI:** 10.3390/antibiotics12010067

**Published:** 2022-12-30

**Authors:** Jinzhou Ye, Xinhai Chen

**Affiliations:** Institute of Infectious Diseases, Shenzhen Bay Laboratory, Shenzhen 518132, China

**Keywords:** antibiotic resistance, new antibiotics, antibiotic efficacy, bacteriophages, vaccine, antibody

## Abstract

Infections caused by antibiotic-resistant bacteria (ARB) are one of the major global health challenges of our time. In addition to developing new antibiotics to combat ARB, sensitizing ARB, or pursuing alternatives to existing antibiotics are promising options to counter antibiotic resistance. This review compiles the most promising anti-ARB strategies currently under development. These strategies include the following: (i) discovery of novel antibiotics by modification of existing antibiotics, screening of small-molecule libraries, or exploration of peculiar places; (ii) improvement in the efficacy of existing antibiotics through metabolic stimulation or by loading a novel, more efficient delivery systems; (iii) development of alternatives to conventional antibiotics such as bacteriophages and their encoded endolysins, anti-biofilm drugs, probiotics, nanomaterials, vaccines, and antibody therapies. Clinical or preclinical studies show that these treatments possess great potential against ARB. Some anti-ARB products are expected to become commercially available in the near future.

## 1. Introduction

Since the discovery of penicillin, the first antibiotic, by Alexander Fleming in 1928, bacterial infectious diseases have ceased to be the leading cause of death worldwide, and the average human life expectancy has almost doubled [1]. However, antibiotic resistance has quickly emerged in numerous clinical bacteria, compromising the initially overwhelming effectiveness of antibiotics. Furthermore, the overuse and misuse of antibiotics have exacerbated this problem of resistance. In 2017, the World Health Organization published a list of twelve bacteria that were of concern as they were all resistant to a notable number of currently marketed antibiotics [2]. These bacteria included the following: *Acinetobacter baumannii* (carbapenem-resistant), *Pseudomonas aeruginosa* (carbapenem-resistant), *Enterobacteriaceae* (carbapenem-resistant, extended-spectrum β-lactamase), *Enterococcus faecium* (vancomycin-resistant), *Staphylococcus aureus* (methicillin-resistant), *Helicobacter pylori* (clarithromycin-resistant), *Campylobacter* spp. (fluoroquinolone-resistant); *Salmonellae* (fluoroquinolone-resistant) and *Neisseria gonorrhoeae* (cephalosporin-resistant, fluoroquinolone-resistant), *Streptococcus pneumoniae* (penicillin-non-susceptible), *Haemophilus influenzae* (ampicillin-resistant), and *Shigella* spp. (fluoroquinolone-resistant). Six of these are common nosocomial pathogens (*E. faecium*, *S. aureus*, *Klebsiella pneumoniae*, *A. baumannii*, *P. aeruginosa*, and *Enterobacter* spp., termed ESKAPE) that often escape the lethal action of antibiotics, as highlighted by the Infectious Diseases Society of America (IDSA) as representative paradigms of pathogenesis, transmission, and resistance [3].

Antibiotic and antimicrobial stewardship is indispensable for combating antibiotic resistance. Netherlands and Sweden, where antibiotic stewardship has been applied in the outpatient setting, are the countries that have the lowest rates of antibiotic resistance in Europe [4]. In England, the reduction in antibiotic prescriptions has greatly attenuated an already increasing incidence of antimicrobial resistance in the subsequently identified *E. coli* bloodstream infections [5]. A systematic review reported that ASPs (antibiotic stewardship programs) could reduce the use of antibiotics, antibiotic costs, treatment duration, and local antibiotic resistance rate without adversely affecting the mortality of patients who require ICU [6]. However, certain limitations still exist that hinder the accurate implementation of antibiotic stewardship. In fear of inadequate coverage of the causative pathogen, physicians empirically prescribe broad-spectrum antibiotics to their patients. As a result, this treatment usually lasts too long or is too broad [7]. Moreover, the high management cost and the patient’s unwillingness to pay hospitalization expenses limit the applicability of antibiotic stewardship in low- and middle-income countries [4].

Many scientists worldwide are now focusing on the development of solutions to combat antibiotic-resistant bacteria (ARB) as a means to prevent the situation of effective antibiotics from becoming clinically unavailable in the future. This review discusses the recent advances in strategies to combat the emergence of ARB based on the literature reporting meritorious chemical, microbiological, and immunological techniques (Figure 1).

## 2. Discovery of New Antibiotics

Most large pharmaceutical companies have scaled back investments in the research and development of new antibiotics. However, research groups at a hospital or academic level outside the industry are still working on feeding the pipeline that targets ARB.

### 2.1. Modifying Old Antibiotics

Modifying the basic chemical structure of an already existing antibiotic may circumvent the resistance mechanisms developed against it. Omadacycline, a semisynthetic tetracycline derivative, has modifications at the C-7 and C-9 positions of the tetracycline D-ring (Figure 2), enabling it to overcome common tetracycline resistance mechanisms, including tetracycline-specific efflux pumps and ribosomal protection [8]. In 2018, the United States approved omadacycline use for treating community-acquired bacterial pneumonia and acute bacterial skin infections. Cefiderocol is a Food and Drug Administration (FDA)-approved cephalosporin-derived antibiotic used to treat carbapenem-resistant *Enterobacterales* and drug-resistant non-fermenting gram-positive bacterial pathogens (GPBPs) [9]. This new antibiotic was synthesized by conjugating a catechol-type siderophore to the typical cephalosporin core (Figure 2), thereby enhancing antibiotic stability against the hydrolytic action of most β-lactamases. Vancomycin and its close chemical relatives are glycopeptide antibiotics that inhibit the growth of gram-positive bacteria by binding to the *L*-Lys-*D*-Ala-*D*-Ala-CO_2_H terminals that need to be cross-linked with pentaglycine to form peptidoglycan, a mesh-like structure surrounding the cytoplasmic membrane that maintains cell shape and protects cells from bursting due to turgor pressure [2,10,11]. Vancomycin is widely used to treat infections caused by *E. faecium* or methicillin-resistant *S. aureus* (MRSA) but has been rendered ineffective as vancomycin-resistant strains are now quite commonly found in hospitals and communities. To overcome vancomycin resistance, the Boger laboratory at the Scripps Research Institute has spent over ten years modifying the basic structure of vancomycin and finally generated a series of vancomycin synthetic analog molecules (also called maxamycins group) (Figure 2) that have notably bactericidal activities against vancomycin-resistant *E. faecium* (VRE) and *S. aureus* (VRSA) isolates [12,13,14]. To produce quantities necessary for preclinical evaluation, this research group has developed a scalable atroposelective total synthesis process of vancomycin analogs that noticeably reduces the number of steps required and enhances the overall yield [15]. This work has, thus, been considered a breakthrough toward the development of next-generation glycopeptide antibiotics with improved activity against vancomycin-resistant bacteria [16].

### 2.2. Identifying New Antibiotics from Small-Molecule Libraries

A small-molecule library typically contains more than 10,000 synthetically produced chemical compounds, which differ in appendages and in their molecular skeleton [17]. Considering the estimation that the probability for a new molecule to become an available medication for patients is only 1:10,000, employing large chemical libraries can increase the chance of discovering new lead antibiotics (Figure 2). One such molecule (compound 1771) was identified by screening 167,405 compounds from small molecule libraries to possess growth-inhibiting properties against antibiotic-resistant GPBPs (such as MRSA and VRE) but not against gram-negative bacterial pathogens (GNBPs) [18]. The compound 1771 targets lipoteichoic acid (LTA) synthase (LtaS), an important enzyme for the growth of gram-positive bacteria. In addition, two anti-MRSA and -VRE compounds (HSGN-94 and HSGN-189) were recently identified by employing a different chemical library to share an aryl-substituted 1,3,4-oxadiazolyl unit with compound 1771 [19,20,21], confirming that the LTA biosynthesis pathway is an essential target for the development of antibiotics against GPBPs. Moreover, some inhibitors targeting wall teichoic acid (WTA) and peptidoglycan pathways are being developed to combat GPBPs. Single deletion of the first two genes (*tagO* and *tagA*) of the WTA biosynthetic pathway results in bacteria lacking WTA polymers while still being viable [22]. However, most of the downstream genes in WTA backbone biosynthesis are essential, and their deletion leads to a lethal phenotype, possibly due to the accumulation of toxic intermediates in the cell or depletion of cellular pools of undecaprenyl phosphate, which is indispensable for the peptidoglycan biosynthesis. Notably, several library screening studies reported that TarG (also known as TagG), a subunit of a two-component ABC transporter that facilitates the translocation of WTA precursors across the membrane, could be a druggable target since all identified compounds could block TarG [23,24,25]. Unfortunately, TarG inhibitors suffered from a relatively high resistance frequency and their inhibitory activities were inconsistent among different isolates [21,25,26]. The peptidoglycan backbone is catalyzed by penicillin-binding proteins (PBPs). The β-lactam antibiotics act by blocking PBPs interaction with the acyl-D-Ala-D-Ala part of the peptidoglycan [27]. MRSA produces a modified penicillin-binding protein called PBP2a, which is insensitive to β-lactam antibiotic binding [28]. The computational screening of 1.2 million drug-like compounds from the ZINC database recently revealed two kinds of PBP2a inhibitors that could protect mice against MRSA and VRSA infection [29,30,31].

The outer membrane of gram-negative bacteria consists of lipopolysaccharide (LPS), outer membrane proteins (OMPs), and lipoproteins, all of which are required for bacterial viability and constitute the major obstacle to the effectiveness of current antibiotics [32]. The essential pathways accounting for the transportation of outer membrane components are Lpt, Bam, Lol, and Sec [33]. Studies exploring antibiotic-like inhibitors targeting these pathways through chemical libraries have been conducted. A clinical antibiotic candidate called murepavadin was discovered by iterative cycles of peptidomimetic library synthesis and screening for improved antimicrobial activity [34,35]. Murepavadin specifically binds to LptD in *P. aeruginosa* and has recently completed a successfully phase-II clinical trial in patients with life-threatening *Pseudomonas* lung infections (clinical trial identifier NCT02096328). Two inhibitors, JB-95 and MRL-494, targeting BamA have been discovered by compound screening [36,37]. BamA is a large β-barrel conserved protein in gram-negative bacteria, implying that BamA inhibitors could have a broad bactericidal activity. The ABC transporter LolCDE captures mature lipoproteins from the inner membrane, which are then transported by LolA to LolB where they are finally translocated into the outer membrane [38]. Many inhibitors of the LolCDE complex have been identified in specific library assays but their development into commercial antibiotics was not possible due to the lack of appropriate physical properties and antimicrobial spectrum required for clinical use [39,40,41,42]. The small-molecule compound MAC13243 presented a potent activity against *Escherichia coli* by directly inhibiting LolA and MreB [43,44]. Antibiotic treatments usually lead to microbiome imbalance, whereas MAC13243 had only a limited impact on the commensal gut microbiota [45], exhibiting a strong specificity. Sec machinery provides a major pathway of protein translocation from the cytosol across the cytoplasmic membrane in both gram-negative and -positive bacteria [46]. One component of this pathway is SecA, which is a widely conserved protein and has been utilized as a druggable target. Several reports indicated that SecA inhibitors (such as SCA107, thiouracil derivatives, and HTS-12302 derivatives) are promising broad-spectrum antimicrobials that could increase membrane permeability by inhibiting the synthesis of membrane proteins, reduce bacterial pathogenicity by suppressing virulence factor production, and overcome multidrug resistance by blocking efflux pumps [47,48,49].

### 2.3. Searching for New Antibiotics from Peculiar Sources

The traditional approach of screening secondary metabolites from soil actinomycetes has led to the discovery of most antibiotics in use today. However, repetitive utilization of this methodology results in the disappointing outcome that most isolated active compounds are already existing antibiotics or their analogs [50]. Thus, many researchers are starting to search for new antibiotics from peculiar sources that include marine samples (invertebrates or algae), invertebrate organisms (such as insects), and microbiomes (Figure 2). Four natural compounds purified from the marine fungal *Stachybotrys* species MF347, have shown anti-MRSA activity [51]. Furthermore, a bactericidal compound MC21-A was isolated from the methanol extract of the marine bacterium *Pseudoalteromonas phenolica* sp. O-BC30T [51] and was effective against MRSA and VRE, likely through the permeabilization of the cell membrane. Thanatin is an insect-derived antimicrobial peptide (AMP) that shows potent activity against various drug-resistant GNBPs and GPBPs [52]. Natural AMPs have intrinsic disadvantages in terms of toxicity, low stability, and high cost. Thanatin variants acquired via engineering had greater tolerance to host proteinases, were non-toxic to human cells, and displayed systemic *in vivo* efficacy against a wide range of drug-resistant pathogens [53,54,55,56,57]. These studies strongly suggest a high potential of thanatin-based products in overcoming the above-mentioned obstacles on the way to clinical success [58]. Lugdunin, a novel thiazolidine-containing cyclic peptide antibiotic produced by nasal *Staphylococcus lugdunensis* strains, has exhibited bactericidal properties against MRSA and VRE, effectiveness in animal models, and low risk in terms of resistance development in *S. aureus* [59], showcasing that human microbiota can be used as a source for mining new antibiotics. Since the majority of microorganisms in nature with a promising potential to be a source of new antibiotics cannot be cultured in artificial growth media, the Lewis laboratory at Northeastern University has developed several methods to address this issue. The group successfully recovered approximately 50% of cells from soil samples by cultivating them in their natural environment or using specific growth factors [60]. An extract termed teixobactin from a new species of β-proteobacteria provisionally named *Eleftheria terrae* showed good activity against GPBPs and *Mycobacterium tuberculosis*. Further mechanistic studies revealed that teixobactin contributes to membrane disruption by concentrating the long hydrophobic tails of lipid II within the supramolecular structure using its C-terminal headgroup to bind the pyrophosphate-sugar moiety of lipid II and N-terminus to coordinate the pyrophosphate of another lipid II molecule [61]. This novel antibacterial mechanism can be utilized to combat current antibiotic resistance.

## 3. Improved Efficacy of Existing Antibiotics

### 3.1. Metabolism Stimulation of Bacterial Pathogens

The emergence of antibiotic resistance predominantly results from mutations in antibiotic-target genes or the transfer of antibiotic-resistant genes between bacterial pathogens [62]. The past decade of research has revealed that bacterial metabolism can contribute to antibiotic resistance [63]. Antibiotic treatment dramatically changes the metabolic state of bacteria, which, in turn, affects their intrinsic susceptibility to the harmful effects of antibiotics. Moreover, plentiful studies have demonstrated that modulating bacterial metabolism is an extremely valid approach to boosting antibiotic effectiveness [64,65,66,67,68,69,70]. To this end, the following two metabolism-based regulatory strategies are available: (i) the enhancement of metabolic pathways that increase antibiotic susceptibility of bacteria and (ii) the inhibition of metabolic pathways that increase antibiotic resistance. Pang et al. observed that a kanamycin-resistant *Edwardsiella tarda* strain was deficient in L-alanine and glucose compared to the wild-type strain [64]. These metabolites markedly improved kanamycin uptake and toxicity through a TCA cycle activation and proton-motive force enhancement mechanism (Figure 3). The elevated intracellular concentration of kanamycin probably exceeded the resistance level caused by spontaneous suppressor mutations. Similar potentiation strategies were implemented in different studies and consistently concluded that ARB could be controlled by antibiotics in combination with a variety of metabolites from glycolysis, TCA cycle, and amino acid metabolism [65,66,67]. Combining phenotypic screening, metabolic network modeling, and white-box machine learning, research was shown that the limitation of intracellular adenine enhanced the activity of ampicillin, ciprofloxacin, and gentamicin [68]. The limitation of adenine by antibiotic treatment stimulates purine biosynthesis, thereby increasing ATP demand and eliciting an accelerated metabolic rate that contributes to cell death. Although the close inter-relationship between bacterial cell metabolism and antibiotic resistance has been frequently utilized to alter the antibiotic susceptibility of bacteria, understanding bacterial metabolism in terms of antibiotic efficacy is still in its infancy. The Collins laboratory at Massachusetts Institute of Technology has claimed that a complete understanding of the relationship between bacterial metabolism and antibiotic function can soon be leveraged into highly potent and precise antibacterial therapies that can counter numerous defense mechanisms used by bacteria to escape inhibition by current antibiotics [63].

### 3.2. Antibiotic Delivery Systems

The recent development of nanomedicine has enabled the design of new drug delivery systems with an improved therapeutic index for the loaded compounds [71]. In addition, some nanomaterials have direct antibacterial activity, which will be discussed later in the section “Developing alternatives to antibiotics”. Nanoparticle carriers can directly deliver antibiotics to the intracellular environment, where drugs are unable to reach therapeutic levels, and can markedly solve the problem of adverse side effects or toxicity caused by high systemic doses and frequent administrations [72]. Synthesized polymer nanoparticles consisting of covalently attached β-lactams were more effective against MRSA compared to drug-sensitive *S. aureus* [73,74]. Furthermore, these nanoparticles had relatively low toxicity and a pronounced anti-MRSA activity whether administrated systemically or topically [75]. Liposomes are lipid-based nano-delivery systems and have been introduced as drug carriers in the 1970s. Recent major breakthroughs in liposome technology have invigorated the interest in using them as efficient antibiotic delivery systems against ARB [76,77]. An antibiotic-liposome drug, Arikayce, was approved by the FDA in 2018 to treat lung disease caused by a group of bacteria belonging to the *M. avium* complex [78]. Because of the similarities between the liposome structure and the bacterial membrane composition, liposomes can stimulate fusion with the bacterial membrane and deliver a high dose of antibiotics inside bacteria (Figure 3). The liposome-bacterial fusion technology is a novel promising approach to overcoming non-enzymatic antibiotic resistance of clinical *P. aeruginosa* strains [79,80], as their resistance mechanisms are largely associated with low and non-specific permeability of their outer membrane or the presence of efflux pumps or both [81]. Furthermore, the antibiotic resistance associated with enzymatic hydrolysis can be conquered by liposome-encapsulated antibiotics. Liposomes prepared with phosphatidylcholine and cholesterol-protected piperacillin against β-lactamase-induced hydrolysis in *Staphylococcus* sp. [82]. *M. tuberculosis* is capable of infecting and persisting in macrophages. Given that liposomes have an inherent advantage to be internalized by macrophages, rifabutin-entrapped liposomes have been produced to treat chronic infections caused by intracellular *M. tuberculosis* [83]. Biomimetic nano-delivery systems offer a relatively new approach that uses membranes of natural cells to load compounds of interest. Among them, outer membrane vesicles (OMVs) derived from gram-negative bacteria have frequently been used as antibiotic carriers and exhibited a strong potential for antibacterial applications. For instance, *A. baumannii* OMVs loaded with fluoroquinolone antibiotics specifically invaded gram-negative bacterial cells of multidrug-resistant *P. aeruginosa*, *K. pneumoniae*, and enterotoxigenic *E. coli* (ETEC) strains and killed them both in vitro and in vivo [84]. Another study used *E. coli*-derived OMVs to encapsulate rifampicin [85], an antibiotic that is typically used for the treatment of *S. aureus* and *M. tuberculosis* infection but not for gram-negative bacteria due to their double-membrane structure. The group reported that rifampicin-loaded OMVs greatly increased the intracellular concentration of rifampicin in *E. coli*, but not in *S. aureus*. When administered to mice, those OMVs elevated the survival rate of infected specimens and reduced the bacterial load in tissues. Additionally, aptamers, small single-stranded oligonucleotides (DNA or RNA) that bind to their specific targets with high affinity and selectivity, can be designed for the delivery of antibiotics. Using a whole bacterium-based procedure of systematic evolution of ligands by exponential enrichment (SELEX), a panel of aptamers specifically binding to *S. aureus* surface was identified and one aptamer SA20hp was selected to test its effect on vancomycin delivery [86,87]. Vancomycin-absorbed capsules were capped with SA20hp gatekeeper molecules to form nanoparticles. Vancomycin was selectively released, killing the pathogen, only when these particles were bound via gatekeeper aptamer sequences to their targets on the *S. aureus* surface. Tests revealed that vancomycin-aptamer composites had a higher anti-*S. aureus* activity than vancomycin alone.

## 4. Developing Alternatives to Antibiotics

Scientists always suggest that the use of antibiotics should be prudent. However, the definition of the term “prudent use” is nebulous. Determining which antibiotics are appropriate or what precise dosage needs to be given to patients remains a challenge for clinicians, even according to the standard medication guidelines. Nonetheless, the key to being prudent about using antibiotics is the use and development of effective alternatives to antibiotics. Successful application of antibiotic alternatives can decrease antibiotic use and hinder the emergence of ARB (Figure 4).

### 4.1. Bacteriophages and Their Encoded Endolysins

In terms of treating bacterial infections, bacteriophage (phage) therapies preceded antibiotic treatment [88]. Phages are duplodnaviria viruses that can exclusively lyse bacteria without harming host cells (Figure 4). Since the introduction of broad-spectrum antibiotics in the 1940s, the development of phage therapy has halted. However, with the emergence of ARB, phage therapy is regaining attention. Currently, several products based on phage therapy are commercially available in some Eastern European countries, while a number of clinical or preclinical studies on phage therapy have been conducted in different countries worldwide. One randomized phase 1/2 trial in France and Belgium evaluated the efficacy and tolerability of a cocktail of phages in patients with *P. aeruginosa* wound infection [89]. Unfortunately, due to the highly unexpected low concentration of phages after manufacturing, this phage therapy showed a slower efficiency in decreasing bacterial load than the standard treatment of burn wounds and needs to be further fine-tuned by increasing phage concentration and evaluated in a large sample of participants [89]. Nonetheless, several case reports originating from China and the United States have shown that phage administration with or without antibiotic treatments could protect patients against multidrug-resistant *A. baumannii* infection [90,91,92], demonstrating the potential of phage therapy against ADR infections in appropriate doses. However, phage bacterial resistance and their immunogenicity are two additional unresolved drawbacks of phage therapy.

Phages employ multiple steps to perform their lytic cycle, primarily by utilizing endolysins, the enzymes that lyse the bacteria by degrading peptidoglycan [93]. Endolysins can be designed to have specificity towards different species of GNBPs and GPBPs. A study reported four recombinant endolysins that could effectively lyse one hundred GNBPs, including multidrug-resistant *K. pneumoniae*, *Salmonella*, *P. aeruginosa*, *E. coli*, *A. baumannii*, and *Enterobacter* spp. strains [94]. An endolysin (Ply6A3) directly purified from phages showed high antibacterial activity against *A. baumannii*, *E. coli*, and MRSA [95]. A new anti-MRSA endolysin, rSAL-1, has been developed and tested as a drug (SAL200), which showed no serious adverse effects in phase-1 of clinical trials [96]. SAL200 is the first endolysin-based drug to be approved for treating human skin infections caused by *S. aureus*, including MRSA [97]. This therapy was highly effective on both chronic and recurrent *S. aureus*-related dermatoses without selecting for bacterial resistance after long-term daily treatment. Another endolysin-based drug, Exebacase, has been developed to treat staphylococcal bloodstream infections, by ContraFect Corporation and is currently in phase 3 of clinical studies [98]. During phase 2, with the aid of standard care antibiotics, a single intravenous infusion of this drug could help patients to combat *S. aureus* bacteriemia and endocarditis [99].

### 4.2. Anti-Biofilm Drugs

Biofilms are a well-known problem in the treatment of bacterial infections. They profoundly hinder the penetration of antibiotics and favor the development of resistance. Thus, biofilm-related infections are impossible to treat with conventional antibiotics. The attachment of bacteria to a surface is the first step in the formation of biofilms. Targeting the initial attachment can be a reasonable strategy to prevent biofilm formation. Mannosides are small molecules that target FimH, an adhesion protein of type 1 pili in *E. coli* [100]. The administration of monomeric biphenyl mannosides prevented the formation of uropathogenic *E. coli* biofilm in vitro and blocked the adherence and invasion of this bacteria in animal models [101]. Antigen 43 and curli fibers are two additional types of adhesion molecules mediating the attachment of bacteria on biotic and abiotic surfaces [102]. Two curlicides (FN075 and BibC6) derived from ring-fused 2-pyridones have been developed to inhibit the biosynthesis of curli fibers needed for biofilm formation [103]. Peptides with anti-biofilm activity (anti-biofilm peptides) are new solutions for biofilm inhibition. Anti-biofilm peptides IDR-1018, DJK-5, and DJK-6 inhibited biofilm formation in multiple GNBPs and GPBPs by binding and degrading guanosine tetraphosphate (ppGpp), which is one of the second messenger nucleotides that sense nutrient starvations and contribute to the antibiotic resistance and biofilm formation [104,105]. In addition, the phage-encoded depolymerases have anti-biofilm activity, as demonstrated by their ability to degrade extracellular polymers and related biofilm components [106]. A depolymerase from φAB6 phage digested capsular polysaccharide, inhibited the colonization of *A. baumannii* on the surface of medical devices, and suppressed *A. baumannii* infection in a zebrafish model [107]. Given the specific recognition of bacterial surface components by aptamers, utilizing aptamers as anti-biofilm agents is possible through the specific binding to certain constituents of the bacterial surface. For instance, three DNA aptamers were identified to bind with the flagellum, a crucial motile component that mediates tropism and initial attachment in the process of biofilm formation, with aptamer 3, having the best anti-biofilm activity [108]. Moreover, using aptamers that bind to the common constituents of both bacterial surface and biofilm is another anti-biofilm strategy. The six DNA aptamers that interacted with *S. aureus* biofilm were demonstrated to assist in the eradication of *S. aureus* biofilm when combined with liposomal delivery of vancomycin and rifampicin [109].

### 4.3. Probiotics

Probiotic supplementation is accepted globally as a beneficial health strategy despite the lack of scientific evidence of its alleged effects. Current mechanisms that explain the protective effect of probiotics are limited to the regulation of the immune system, enhancement of the intestinal epithelial barrier, competition with pathogenic bacteria for nutrients, and bacteriocin-mediated interference [110]. The contribution of the microbiome to host defense against pathogen colonization and prevalence is called “colonization resistance” [111]. *Bacillus subtilis*, a common strain of human gut microbiota and component of probiotic formulae, can produce bacitracin that interferes with the cell wall synthesis of MRSA and other GPBPs. However, in a randomized prospective study involving a 5-day treatment in healthcare workers, bacitracin was inferior to mupirocin for suppressing nasal *S. aureus* [112]. Besides bacitracin, *B. subtilis* produces fengycins, which are a family of lipopeptides. Fengycins exhibited a strong ability to block the *S. aureus* quorum-sensing system Agr, which is vital for *S. aureus* gut colonization [113]. These studies suggest a strong potential for using alive *Bacillus* strains as a probiotic therapy to prevent infections caused by *S. aureus* colonization. Besides natural probiotics, engineered or synthetic probiotics are an additional promising option. For example, an engineered probiotic *E. coli* Nissle 1917 equipped with quorum sensing and lysing devices sensed and killed 99% planktonic *P. aeruginosa*, and caused a 90% reduction in biofilm formation in an in vitro system [114]. More importantly, this research group demonstrated that this engineered strain could be applied in animal models to suppress *P. aeruginosa* infection [115]. Although probiotic therapies related to colonization resistance are clinically unavailable yet, the elucidation of mechanisms that are responsible for such resistance can pave the way to the successful development of probiotic treatments.

### 4.4. Antibacterial Nanomaterials

Many nanomaterials have an inherent bactericidal activity through several well-accepted mechanisms, including oxidative stress response, physical disruption, altering bacterial metabolism, protein denaturation, and disrupting DNA replication [116]. Normally, the bacterial cell envelope is negatively charged, and electrostatic interaction can be achieved by designing positively charged nanoparticles. Gold nanoparticles caused bacterial membrane tension and squeezing, resulting in physical damage to the cell envelope and subsequent cell lysis [117]. Other metal nanoparticles, such as silver and copper, could also destroy the bacterial cell wall/membrane [118,119]. Silver nanoparticles are capable of damaging peptidoglycan structure by generating reactive oxygen species (Figure 4) [120]. Copper nanoparticles reduced the expression of a bacterial glucose transporter and the activity of nitrate and nitrite reductases [121]. With selected bacteria as the target, the cationic polymer coating of nanoparticles showed antibacterial activity by destroying bacterial cell plasma membrane, inhibiting bacterial proliferation, and preventing biofilm formation through strong electrostatic interaction with the negatively charged bacterial membrane [122]. These single-element nanoparticles show nontargeted antibacterial activity, while the development of composite nanoparticles may increase specificity and reduce the damage to microbiota. Based on the differences in the cell envelope structure between gram-positive and gram-negative bacteria, gold nanocomposite particles based on amino sugars with a narrow spectrum antibacterial activity and a gram-positive antibacterial action were developed [123]. Graphene oxide-silver (GO-Ag) nanoparticles had differential inhibitory effects on gram-negative *E. coli* and gram-positive *S. aureus* [124]. GO-Ag nanoparticles showed a bacteriostatic effect towards *E. coli* and *S. aureus* by destroying the integrity of bacterial cell walls, and inhibiting the cell division cycle, respectively.

### 4.5. Vaccines

Vaccines are always the first choice to prevent infectious diseases. Compared to the relatively simpler viruses, fewer vaccines are clinically available for bacteria. As of 2019, FDA approved 65 and 32 vaccines to prevent diseases caused by viruses and bacteria, respectively. These bacteria were *B. anthracis*, *M. tuberculosis*, *Vibrio cholerae*, *Clostridium tetani*, *H. influenzae*, *Neisseria meningitidis*, *Yersinia pestis*, *S. pneumoniae*, and *Salmonella typhi*, which do not include the nosocomial ESKAPE pathogens. No vaccine against ESKAPE pathogens is available, though not for lack of trying as demonstrated by dozens of clinical trials that have failed so far. In earlier clinical trials, vaccinating whole cells or cell lysates of pathogens such as *K. pneumoniae* and *S. aureus* exhibited a limited protection with relatively high toxicity [125,126]. StaphVAX, a vaccine that targeted *S. aureus* capsular polysaccharide types 5 and 8 (CP5 and CP8), was assessed in two phase III studies and eventually displayed no benefit for the recipients with bacteremia [127]. The IsdB vaccine V710 also failed in a phase III randomized controlled trial of *S. aureus* infection [128]. Pfizer’s SA4Ag vaccine, targeted multicomponent antigens MntC, ClfA, and both CP5 and CP8, was still insufficient in providing protective immunity against *S. aureus* in the bloodstream and surgical site infections in patients who had undergone spinal surgery [129]. Two LPS-targeted vaccines were unsuccessful in reducing the rate of *P. aeruginosa* colonization and the frequency of *P. aeruginosa* infection in patients with cystic fibrosis [130,131]. The vaccine IC43 consisting of two *P. aeruginosa* OMPs, OprF, and OprI, had no significant effect on invasive *P. aeruginosa* mortality and infections in a phase III study [132]. These clinical studies urge scientists to dissect disappointing results in the hope of finding ways to overcome failures. Recent studies strongly suggest that prior exposures to bacteria by colonization seem to cause the ineffectiveness of vaccines in humans. Such bacteria that are commonly found in human microbiota include *E. faecium*, *S. aureus*, *K. pneumoniae*, and some *Enterobacter* strains [133,134]. The pre-existing human-*S. aureus* interaction jeopardizes immune responses to many staphylococcal antigens [135,136,137,138]. Thus, in humans who have been previously exposed to *S. aureus*, new vaccination with those antigens is incapable of conferring protective immunity. Overcoming this obstacle of prior *S. aureus* exposures can be achieved at least by immunizing the antigen epitopes that only develop protective immunity or that have no immune response during *S. aureus* exposures. Staphylococcal protein A (SpA) is one such antigen that has no specific antibodies in humans and vertebrates after colonization or infection [139,140]. Two improved nontoxigenic SpA variants were developed and showed strong potential to reduce the rate of *S. aureus* bloodstream infection and nasal colonization [141], and now are being further studied.

### 4.6. Antibody Therapies

Antibodies can be used to prevent and treat diseases. In contrast to several of the aforementioned available clinical vaccines, only three FDA-approved antibody therapies for bacterial infections exist, two of which treat *B. anthracis* infection and the other one treats *Clostridium difficile* infection [142]. All of these antibodies consist of human immunoglobulin G (IgG) and recognize secretory toxins. However, many other antibody therapies targeting components of secretory toxins or cell envelopes of GNBPs and GPBPs have failed in clinical trials [142]. As we mentioned above, protective antigen epitopes are important, and most antigens contain both protective and nonprotective epitopes. Thus, the antibodies uniquely recognizing protective epitopes are more likely to have a therapeutic effect on infections. For instance, in previously *S. aureus*-exposed hosts, antibodies elicited from NEAT2 but not the NEAT1 domain of IsdB provided protection against secondary *S. aureus* infection [138]. More importantly, out of three monoclonal antibodies (mAbs) recognizing the NEAT2 domain, only one (mAb 1C1) provided protection under the above infection circumstances because it resisted competition by antibodies induced by previous *S. aureus* exposures. Other factors besides the appropriate paratope, including isotype, antibody titer, half-life, antigen-binding affinity, N-297 glycosylation, downstream effectors (complement C1q, FcγRs), and bacterial evasion strategies, can also impact antibody functions against bacterial infections. A mAb 3F6 recognizing the IgG-binding protein (IBP) nSpA, exerted anti-*S. aureus* activity only under high N-297 galactosylation or afucosylation conditions [143]. The favorable glycosylation enhances the antibody 3F6-IgG interaction with C1q or FcγR, strengthening their opsonophagocytic killing (OPK) of *S. aureus*. Distinct evasion strategies specific to antibodies have been explored in different bacteria. For instance, *S. aureus* can inhibit antibody functions by producing SpA and IgG-degrading V8 proteases [144,145]. SpA captured the Fc region of IgG, keeping it from binding to FcγRs, C1q, and FcRn and resulting in decreased OPK activity and half-life of IgG [146]. Notably, mAb IgG variants with R-QVV and R-DDRVV substitutions in the Fc region could disrupt SpA interference improving therapeutic activity in humanized FcRn mouse models. This study demonstrated the significance of overcoming IgG-specific evasion strategies by modifying the Fc region of IgG. Similar to *S. aureus*, ETEC and *Streptococcus pyogenes* produce IBPs that bind to the Fc region of antibodies [147,148,149,150]. Furthermore, *S. pyogenes* and *P. aeruginosa* secreted IgG-degrading enzymes, which cleaved the hinge region or N-glycosylation of IgG [151,152]. In samples collected from patients with *S. pyogenes*, IgG-Fc was found to bind on the surface IBPs of *S. pyogenes*, along with the detection of cleavage of N-glycosylation of IgG [152,153]. In conclusion, future directions of generating antibody therapies against bacterial infections should simultaneously consider their paratopes that bind with protective epitopes and their modification by engineering to overcome IgG-specific evasion strategies.

Other antibody engineering options for combating bacterial infections have been reported. DSTA4637S is an antibody-antibiotic conjugate that combines a rifampicin-class antibiotic with an engineered human IgG1 that specifically binds to N-acetyl-glucosamine of WTA [154]. This conjugate is designed to kill intracellular reservoirs of *S. aureus*. When DSTA4637S-opsonized *S. aureus* is taken up by host cells, phagolysosomal proteases cleave the linker between antibody and antibiotic releasing the antibiotic in sufficient concentrations to kill the bacteria. MEDI13902 is a bispecific antibody against *P. aeruginosa* PcrV, a protein located at the top of the type III secretory system, and Psl, an extracellular polysaccharide [155]. This antibody therapy is currently in phase II of the clinical trial to evaluate the effectiveness and safety issues in mechanically ventilated patients. Antibody engineering has been widely used in cancer immunotherapy [156] but has only just begun for bacterial infections. Further investigations considering multiple host and bacterial factors are expected to result in promising antibody therapies.

## 5. Future Outlook

Nowadays, due to the growing problem of antibiotic resistance, the development of novel anti-ARB therapeutic strategies has been of great interest. Although certain strategies discussed in this review have shown promising potential in clinical or preclinical studies, the complexity of their structure and physiology may protect bacteria from single treatments. The combination of these strategies and flexibility in their use could hold more promise for future anti-ARB treatments. For instance, in order for the antibody–antibiotic conjugate (DSTA4637S) to kill intracellular reservoirs of *S. aureus,* the binding of the antibody to *S. aureus* is required [154]. However, bacteria that have already survived in the cytosol before DSTA4637S treatment are still hard to be killed due to spatially impermissible interaction between DSTA4637S and those cytosolic bacteria. In this case, antibiotic-encapsulated liposomes or OMVs can be used together with DSTA4637S to kill those intracellular bacteria. Moreover, the antibody in DSTA4637S does not have any actual OPK activity and only serves as a staphylococcal targeting molecule for the antibiotic to kill intracellular *S. aureus*. Antibodies that could both sensitize pathogens enhance the efficacy of antibiotics and exhibit OPK activity would be preferred. For example, MprF-targeting mAb blocks bacterial lipid translocation and can sensitize antibacterial agents [157], indicating that antibodies that recognize membrane proteins seem to be ideal candidates to trigger both OPK activity and sensitization of current antibiotics.

## Figures and Tables

**Figure 1 antibiotics-12-00067-f001:**
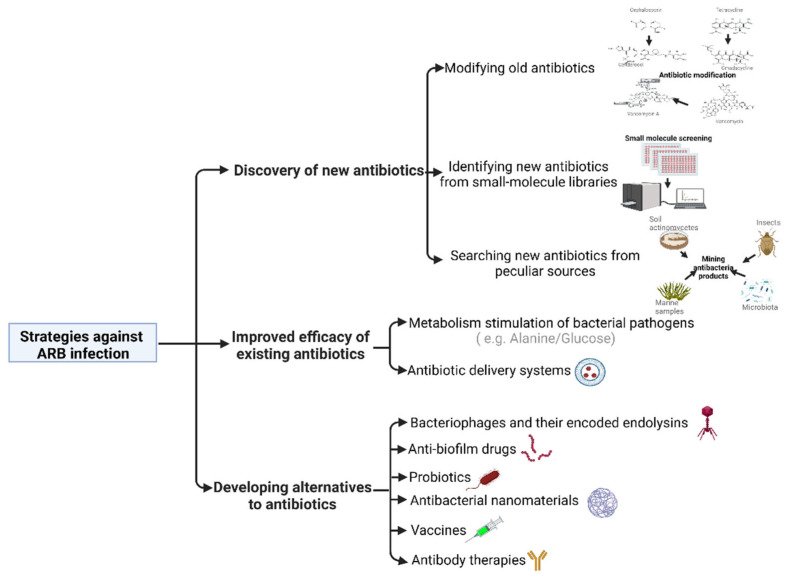
Strategies against antibiotic-resistant bacterial infection.

**Figure 2 antibiotics-12-00067-f002:**
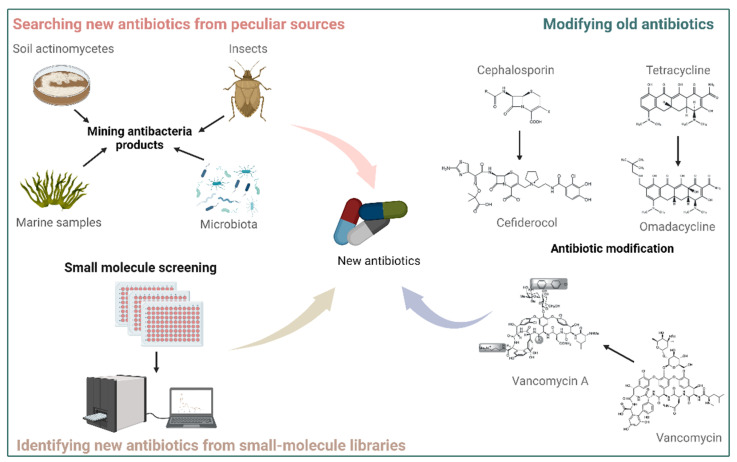
New antibiotics are mined by the following three strategies: modification of old antibiotics, screening of small-molecule libraries, and exploration of peculiar places.

**Figure 3 antibiotics-12-00067-f003:**
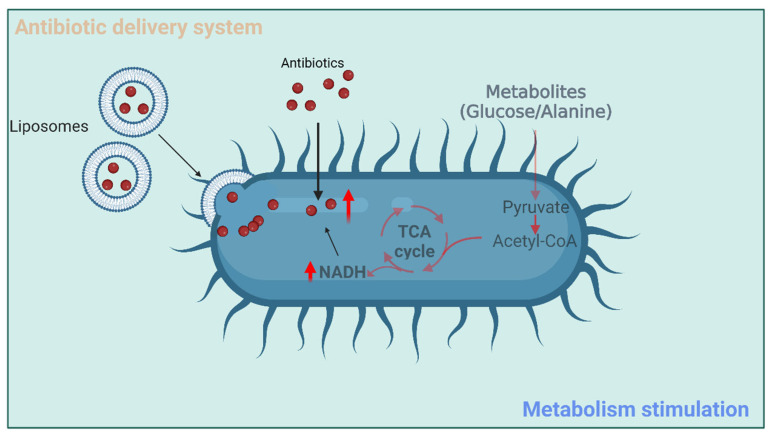
The ways to improve efficacy of current antibiotics.

**Figure 4 antibiotics-12-00067-f004:**
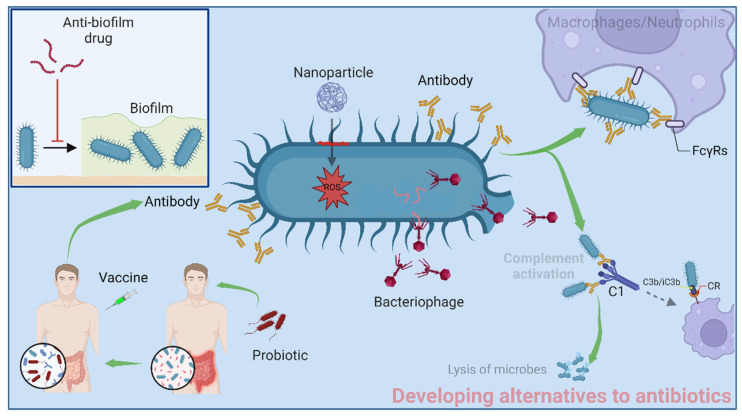
Alternatives to antibiotics.

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
