# Peer review of "Current Promising Strategies against Antibiotic-Resistant Bacterial Infections"

_antibiotics, 2022, doi:10.3390/antibiotics12010067_

Round 1

Reviewer 1 Report

Firstly, I would like to say that this is an excellent article. I enjoyed the topic and the presentation as a whole. In spite of their promising nature, these antibiotic-resistant bacterial infection strategies cannot be implemented immediately.  Would it be possible for the authors to include a paragraph on antibiotic stewardship, particularly in intensive care units? Ideally, these citations should be from the last five years and have a high impact factor.

Author Response

Question: Would it be possible for the authors to include a paragraph on antibiotic stewardship, particularly in intensive care units? Ideally, these citations should be from the last five years and have a high impact factor. 

Answer: Thank you for this good comment. We agree that the antibiotic stewarship has a pivotal position now to reduce the use of antibiotics and have included a paragraph about the antibiotic stewardship in the "Introduction" part.

Reviewer 2 Report

The review covers strategy to combat emergence of antibiotic resistant bacterial infections. The review is important since ARB is considered to be a major threat to public health. The review is well written covering current strategies to tackle ARB ranging from eliminating mature biofilms to progress in related development of vaccines and drug delivery nanosystems. 

The only comment I have is regarding the organization of manuscript. The authors included relevant information in text and readers have to remember everything for comparison. I will recommend the authors to present information related to mentioned strategies in form of figures or tables. This will reduce the length of text and make information comprehensible for reader.

Author Response

Question: The authors included relevant information in text and readers have to remember everything for comparison. I will recommend the authors to present information related to mentioned strategies in form of figures or tables.

Answer: Thanks for the recommendation. Now we have included the Figure 1 to summarize the mentioned strategies.

Round 2

Reviewer 1 Report

Thanks to the author for the thoughtful reply that made this article even better. In my opinion, a good antibiotic stewardship program can lead to reduce antibiotic consumption, expenditures, and antimicrobial resistance, as well as improve in-hospital mortality and adequate antibiotic use. I believe the author did his best, and my questions mostly resolved the issue as well.

Author Response

Thank you! We are very glad you agree our revision.